# Identification, Isolation, and Characterization of Melanocyte Precursor Cells in the Human Limbal Stroma

**DOI:** 10.3390/ijms23073756

**Published:** 2022-03-29

**Authors:** Shen Li, Matthias Zenkel, Friedrich E. Kruse, Andreas Gießl, Ursula Schlötzer-Schrehardt

**Affiliations:** Department of Ophthalmology, University Hospital Erlangen, Friedrich-Alexander-University Erlangen-Nürnberg, Schwabachanlage 6, D-91054 Erlangen, Germany; shen.li@uk-erlangen.de (S.L.); matthias.zenkel@uk-erlangen.de (M.Z.); friedrich.kruse@uk-erlangen.de (F.E.K.); andreas.giessl@uk-erlangen.de (A.G.)

**Keywords:** limbal stem cells, limbal stem cell niche, melanocytes, limbal stroma, magnetic-activated cell sorting, cultivation, laminin 511-E8

## Abstract

Given their vital role in the homeostasis of the limbal stem cell niche, limbal melanocytes have emerged as promising candidates for tissue engineering applications. This study aimed to isolate and characterize a population of melanocyte precursors in the limbal stroma, compared with melanocytes originating from the limbal epithelium, using magnetic-activated cell sorting (MACS) with positive (CD117/c-Kit microbeads) or negative (CD326/EpCAM or anti-fibroblast microbeads) selection approaches. Both approaches enabled fast and easy isolation and cultivation of pure limbal epithelial and stromal melanocyte populations, which differed in phenotype and gene expression, but exhibited similar functional properties regarding proliferative potential, pigmentation, and support of clonal growth of limbal epithelial stem/progenitor cells (LEPCs). In both melanocyte populations, limbus-specific matrix (laminin 511-E8) and soluble factors (LEPC-derived conditioned medium) stimulated melanocyte adhesion, dendrite formation, melanogenesis, and expression of genes involved in UV protection and immune regulation. The findings provided not only a novel protocol for the enrichment of pure melanocyte populations from limbal tissue applying easy-to-use MACS technology, but also identified a population of stromal melanocyte precursors, which may serve as a reservoir for the replacement of damaged epithelial melanocytes and an alternative resource for tissue engineering applications.

## 1. Introduction

Corneal transparency and visual function rely on a healthy population of epithelial stem/progenitor cells that reside in a specialized and highly regulated niche at the corneoscleral limbus, which is clinically identified as palisades of Vogt [1,2,3]. In this specific microenvironment, stem cell phenotype and function are regulated by complex interactions with non-epithelial niche cells, extracellular matrix components, blood vessels, and nerves [4,5,6]. Melanocytes represent an essential component of the limbal stem cell niche [7,8,9,10,11,12]. They are located within the limbal basal epithelium, and are known to protect limbal epithelial stem/progenitor cells (LEPCs) from UV damage by transferring melanosomes via multiple dendritic cell processes. Our previous studies provided evidence that limbal melanocytes are not only professional melanin-producing cells but exert various non-canonical functions in limbal niche homeostasis by regulating LEPC maintenance, immune responses, and angiostasis [13]. Given the vital role of melanocytes in LEPC and niche protection, continuous crosstalk between LEPCs and melanocytes is required to maintain a ratio of approximately 8–10:1 of LEPCs to melanocytes in the basal epithelial layer [13]. It is, however, not clear whether the existing melanocytes can divide or whether they arise from precursor cells in the limbal stroma to replace damaged melanocytes within the basal epithelium. Despite these unresolved issues, limbal melanocytes have emerged as interesting candidates for novel tissue engineering strategies in ocular surface disorders due to their potent regulatory, immunomodulatory, and anti-angiogenic properties [13,14]. The development of such novel approaches, however, requires reliable protocols for the efficient isolation and cultivation of limbal melanocytes from donor corneas.

Since the first long-term culture of pure melanocytes was established in 1982 [15], it was realized that the cultivation efficiency relies on the stimulation of melanocyte proliferation and complete elimination of contaminating cells, mainly keratinocytes and fibroblasts. Phorbol 12-myristate 13-acetate (PMA) has been commonly used to promote the proliferation of melanocytes derived from skin, uvea, and conjunctiva and suppress the growth of contaminating keratinocytes, but was gradually replaced by mitogens and growth factors [15,16,17,18]. The antibiotic geneticin, an inhibitor of protein synthesis, was widely used to deplete contaminating fibroblasts from melanocyte cultures through its cytotoxic effect on fast-dividing cells [19,20], and has also been applied to the long-term cultivation of limbal melanocytes [14,21]. It became, however, increasingly clear that this destructive procedure is not only time-consuming but may cause adverse effects on melanocytes by leading to latent phenotypical and functional cellular changes [20,22]. To obviate these problems, a direct method of melanocyte isolation from tissues using fluorescence-activated cell sorting (FACS) for the melanocyte surface marker CD117 (c-Kit) was introduced to enrich both epidermal and limbal melanocytes [22,23]. FACS technology is, however, also relatively time-consuming and requires expensive equipment, which is mostly provided only at core facilities.

Magnetic-activated cell sorting (MACS) has been widely used for cell isolation due to its less time-consuming and less expensive equipment requirements [24,25]. Therefore, we established an alternative method based on easy-to-use MACS technology for the direct isolation, enrichment, and cultivation of melanocytes from limbal epithelial and stromal tissues, which proved superior to conventional culture methods. Both positive and negative selection approaches generated pure epithelial and stromal melanocyte populations, which differed by phenotype and gene expression patterns, but exhibited similar functional properties. We further demonstrated the effect of limbus-specific matrix and soluble factors on melanocyte adhesion, dendrite formation, melanogenesis, and the expression of genes involved in UV protection and immune regulation. The findings provide not only a simple and fast protocol for the efficient isolation and enrichment of pure melanocyte populations from limbal tissue, but also provide evidence of a stromal melanocyte precursor population, which may serve as a reservoir for epithelial melanocytes and an alternative resource for tissue engineering approaches.

## 2. Results

### 2.1. Localization of Two Melanocyte Populations at the Human Limbus In Situ

Immunohistochemical analyses of corneoscleral tissue sections using antibodies against the melanocyte marker Melan-A showed that melanocytes were localized within the basal limbal epithelium as previously described [13]. In addition, small Melan-A positive cells, partly with a spindle-shaped phenotype, were observed in the anteriormost limbal stroma beneath the basal epithelial cell layer stained for integrin α6 (ITGA6) (Figure 1).

### 2.2. Isolation and Enrichment of Limbal Epithelial and Stromal Melanocytes

As outlined in Figure 2A, the limbal epithelium was enzymatically separated from the underlying stroma using dispase II. Single-cell suspensions were generated from both the epithelial and stromal tissues by trypsin/EDTA and collagenase A digestion, respectively.

Using the conventional cultivation method, pure cultures of epithelial and stromal melanocytes required sequential treatment with geneticin to remove contaminating fibroblasts, amounting to a total cultivation period of about 3 months [21].

By using MACS technology, epithelial and stromal melanocytes taken from mixed cell populations were enriched either by positive selection using CD117 (c-Kit) MicroBeads, or by negative selection using CD326 (EpCAM) and Anti-Fibroblast MicroBeads to remove contaminating epithelial cells and fibroblasts from epithelial and stromal cell populations (Figure 2A). The fraction of CD117^+^ cells in mixed cell populations varied from 1 to 7%, most likely dependent on donor age, tissue quality, and duration of organ culture. Both negative selection approaches required another 1 to 2 (epithelial fraction) or 2 to 3 (stromal fraction) MACS purification steps with Anti-Fibroblast MicroBeads to completely eliminate any remaining fibroblasts. Thus, relatively pure cultures of epithelial or stromal melanocytes were obtained after 1 to 3 passages. Overall, both positive and negative selection approaches were equally effective and reduced cultivation time until confluency from about three to one month. Since, however, the negative selection protocols involved less manipulation of the target cells, these protocols were used for further experiments.

MACS technology not only significantly reduced culture time but also improved the viability of cultivated melanocytes. During conventional geneticin treatment (200 µg/mL), many cells of both epithelial and stromal fractions died, producing abundant detrimental cellular debris (Figure 2B). MTT viability assay confirmed that geneticin at concentrations of 100–400 µg/mL decreased the viability of melanocytes by 38–47% after 24 h compared to untreated control cells, but became less harmful after 48 h and 72 h of treatment (Figure 2C). A concentration of 400 µg/mL, however, caused a significant reduction in cell viability at all time points.

Flow cytometry analysis showed enrichment efficiency of melanocytes by sequential negative selection procedures (Figure 2D). In the epithelial fraction, the proportion of Melan-A^+^ cells increased from 0.7% before MACS purification to 11.0% after one purification step (P0) and to 93.0% after two additional purification steps (P2), which corresponded to a pure population of epithelial melanocytes (95.1% at P4). The stromal fraction could be enriched for Melan-A^+^ cells from 1.1% before MACS separation to 19.3% after the first purification step (P0) and to 57.2% after two additional cycles (P2). To obtain a relatively pure population of stromal melanocytes (87.1% at P4), another purification step with Anti-fibroblast MicroBeads would be needed. Since CD117 is not specifically expressed on melanocytes but also on other cell types, including hematopoetic and mesenchymal stem cells, the percentages of Melan-A^+^/CD117^+^ cells were slightly lower than that of all Melan-A^+^ cells.

### 2.3. Comparative Characterization of Limbal Epithelial and Stromal Melanocytes

Limbal epithelial and stromal melanocyte populations revealed distinct phenotypes on phase-contrast microscopy showing rather multi-dendritic, flattened epithelial melanocytes and fairly bipolar, spindle-shaped stromal melanocytes (Figure 3A).

Immunocytochemistry using antibodies against known melanocyte markers [2] showed that both epithelial and stromal melanocytes were positive for Melan-A (MART-1), CD117/c-Kit, the premelanosome marker HMB-45 (human melanoma black-45; PMEL [premelanosome protein]), the transcription factors MiTF (microphthalmia-associated transcription factor) and Sox10 (SRY-box transcription factor 10) as well as the proliferation marker Ki-67 (Figure 3B). While the tyrosinase-related protein TRP1 was similarly expressed in both populations (not shown), TRP2 was more prominent in epithelial than stromal cells. This was also observed for the cell-cell adhesion molecule E-cadherin, which showed increased immunopositivity in epithelial compared to stromal melanocytes. Clear differences were observed in the staining patterns for ß-catenin, an important component of the Wnt signaling pathway, which showed cytoplasmic and cell surface staining in epithelial melanocytes but nuclear staining in stromal melanocytes. The pluripotent stem cell marker Sox2 was similarly expressed in a small percentage of cells of both populations. The neural crest stem cell marker nestin was predominantly expressed in a subset of Melan-A^dim^ stromal melanocytes, but more uniformly expressed in Melan-A^+^ epithelial melanocytes. In contrast, nerve growth factor receptor NGFR/p75^NTR^, another neural crest stem cell marker, was exclusively expressed in individual cells of the epithelial melanocyte population (not shown).

Real-time RT-PCR largely confirmed and expanded the protein expression data through the inclusion of additional melanocyte-associated genes (Figure 4A). Epithelial and stromal melanocytes showed no significant differences in expression levels of melanocyte markers MLANA (MART-1; Melan-A), TYRP1 (tyrosinase-related protein 1), and TYR (tyrosinase), the melanosome/exosome markers PMEL (premelanosome protein; HMB-45) and CD63 (CD63 antigen), the melanocyte precursor markers MITF (microphthalmia-associated transcription factor) and KIT (c-Kit/CD117), the neural crest cell marker SOX10 (SRY-Box transcription factor 10), the pluripotent stem cell markers POU5F1 (POU class 5 homeobox 1; Oct-4), SOX2 (SRY-Box transcription factor 2) and ABCB5 (ATP binding cassette subfamily B member 5), the regulatory genes MC1R (melanocortin 1 receptor), CTNNB1 (catenin beta 1), KITLG (Kit ligand; SCF, Stem cell factor) and CXCR4 (stromal cell-derived factor 1 receptor), as well as the proliferation marker MKI67 (Ki-67). Epithelial melanocytes showed significantly higher expression levels of TYRP2 (tyrosinase-related protein 2), NES (nestin), NGFR (nerve growth factor receptor; p75^NTR^), CXCL12 (C-X-C Motif Chemokine Ligand 12; SDF-1, stromal cell-derived factor 1) and the differentiation marker CDH1 (E-cadherin).

Epithelial and stromal melanocytes did, however, not differ significantly in their proliferative potential (Figure 4B) or melanin content (Figure 4C). Using melanocytes as feeder layers for LEPCs, both epithelial and stromal melanocytes supported the clonal growth of LEPCs. Colony-forming efficiency and colony growth area exhibited no significant differences between the two feeder layers indicating that both populations may be suitable for supporting LEPCs in tissue engineering approaches (Figure 4D).

### 2.4. Influence of Limbal Niche Factors on Melanocyte Phenotype

Given the comparable functional properties of epithelial and stromal melanocytes, we hypothesized that their differences in phenotype are determined by their different microenvironments, such as extracellular matrix components and soluble factors secreted by neighboring cells. Whereas epithelial melanocytes may be mainly regulated by niche-specific basement membrane components such as LN-511 and soluble factors secreted by LEPCs [21], stromal melanocytes may be influenced by stromal matrix components such as collagen type I and factors derived from limbal mesenchymal stromal cells (LMSCs).

Compared to uncoated tissue culture plastic, coating with recombinant LN-511-E8 supported the formation of a flattened phenotype of both melanocyte populations indicating increased cell attachment (Figure 5A). In contrast, collagen type I supported the formation of a spindle-shaped phenotype in both populations, indicating decreased cell attachment (Figure 5A). The proliferative capacity of both epithelial and stromal melanocytes was significantly enhanced by LN-511-E8 and reduced by collagen type I (Figure 5B). In contrast, LN-511-E8 and collagen type I did not alter the melanin production of the two populations (Figure 5C). By LN-511-E8, the mRNA levels of CXCL12 and SOX2 were significantly upregulated in epithelial melanocytes, whereas levels of NGFR and ABCB5 were upregulated in stromal melanocytes. Epithelial melanocytes further showed a significant downregulation of MITF and KIT (Figure 5D). Collagen type I had little effect on mRNA expression levels of both melanocyte populations, except for a significant downregulation of CXCL12 and CXCR4 mRNA in epithelial melanocytes (Figure 5D). Expression levels of all other genes analyzed were not affected by matrix components (not shown).

A conditioned medium derived from LEPC induced the formation of small spiny dendriform cell processes by both melanocyte populations, which connected to neighboring cells, whereas a conditioned medium derived from LMSC promoted a spindle-shaped, bipolar phenotype (Figure 6A). Conditioned medium from LMSC but not from LEPC increased proliferation of both melanocyte populations (Figure 6B). Both LEPC and LMSC conditioned media had, however, no significant effect on melanocyte pigmentation (Figure 6C). Under the influence of LEPC conditioned medium, mRNA levels of TYRP1, TYRP2, and PMEL increased, and levels of NGFR decreased in epithelial melanocytes. In contrast, stromal melanocytes downregulated mRNA levels of SOX10, KIT, NES, ABCB5, and CXCR4 (Figure 6D). Effects of LMSC conditioned medium were less pronounced, and included upregulation of MLANA and TYRP2 in epithelial melanocytes and downregulation of SOX2 and KITLG in stromal melanocytes (Figure 6D). Expression levels of all other genes analyzed were not affected by soluble factors (not shown).

## 3. Discussion

Distinct melanocyte stem and precursor cell populations have been reported and characterized in adult mice and humans, particularly in hair follicles and extrafollicular dermis, which can differentiate into mature melanocytes in vitro [26,27,28]. In a three-dimensional skin equivalent model, melanocyte precursor cells differentiated into HMB-45-positive melanocytes, which migrated from the dermis to the epidermis and aligned among the basal layer keratinocytes by acquiring E-cadherin expression [29]. The renewal of mature melanocytes in situ is also presumed to occur from these progenitor cell populations, whereby the differentiation into melanocytes is governed by local environmental factors. The melanocyte precursor cell population was described as small, bipolar, and weakly pigmented, whereas the differentiated melanocyte cell population was characterized as larger, dendritic, and intensely pigmented [26,28].

To the best of our knowledge, this is the first report showing the presence of subepithelial melanocytes in the anteriormost subepithelial stroma at the human limbus. The melanocytes were identified by the expression of the specific melanocyte marker Melan-A in situ. They were further characterized by immunocytochemistry, qPCR, and functional assays in vitro through direct comparison with intraepithelial melanocytes isolated from the same donor corneas. Stromal and epithelial melanocytes exhibited distinct phenotypes, i.e., bipolar spindle-shaped stromal cells and multi-dendritic flattened epithelial cells, which are compatible with precursor and mature melanocyte populations as described in skin and hair follicles. Compared with limbal epithelial melanocytes, stromal melanocytes showed reduced expression of the melanocyte differentiation marker TYRP2 (TRP2), the differentiation marker CDH1 (E-cadherin), the neural crest markers NES (nestin) and NGFR (p75^NTR^), and the chemokine CXCL12 both on the mRNA and protein level. A lack of NGFR/p75^NTR^ and CXCL12 expression by stromal melanocytes was the most significant distinguishing feature between both populations. Stromal melanocytes further exhibited nuclear staining for ß-catenin, indicating activation of Wnt signaling, which plays a crucial role in melanocyte proliferation, migration, and differentiation [30]. However, their proliferative potential, melanin content, and colony-forming efficiency of LEPCs was similar to that of epithelial melanocytes. The functional significance of this stromal melanocyte population remains unclear, but it is conceivable that they represent a migratory precursor population and a reservoir for replacing damaged melanocytes in the limbal epithelium. Given the vital role of epithelial melanocytes in UV protection and homeostasis of the limbal stem cell niche [13], there is a high probability of a precursor cell pool in a well-protected area of the limbus.

To confirm that the local microenvironment modulates melanocyte phenotype and function, we exposed both melanocyte populations to limbal epithelial and stromal extrinsic cues. Whereas the limbus-specific basement membrane component LN-511-E8 stimulated cell attachment, spreading, and proliferation, the stromal matrix component collagen type I decreased attachment and proliferation of both melanocyte populations. LN-511-E8 also upregulated mRNA expression levels of NGFR/p75^NTR^, the chemokine CXCL12, and its receptor CXCR4, as well as levels of stem cell markers SOX2 and ABCB5 in either epithelial or stromal melanocytes, and downregulated levels of melanocyte precursor markers MITF and KIT in epithelial melanocytes. Expression levels of CXCL12 and CXCR4 in epithelial melanocytes were downregulated by collagen type I. LEPC-derived conditioned media stimulated dendrite formation in both melanocyte populations and induced expression of genes involved in melanogenesis, i.e., TYRP1, TYRP2, and PMEL, in epithelial melanocytes. This is paralleled by a small but statistically insignificant increase in melanin content after treatment with LEPC conditioned medium, which may be explained by a delay between gene transcription and melanin production. In stromal melanocytes, expression levels of stem cell and precursor markers, i.e., SOX10, NES, KIT, and ABCB5, were downregulated by LEPC conditioned medium. LMSC-derived conditioned medium stimulated the formation of a spindle-shaped phenotype and proliferation rates of both melanocyte populations; it also induced expression of genes involved in melanosome biogenesis, i.e., MLANA and TYRP2, in epithelial melanocytes and downregulated SOX2 and KITLG in stromal melanocytes. These observations suggest that contact of melanocytes with the limbal basement membrane induces their attachment and increased expression of NGFR/p75^NTR^ and CXCL12/CXCR4, whereas LEPC-derived soluble factors induce dendrite formation and melanogenesis. In practical use, conditioned medium from LMSC may serve to increase the proliferation and expansion of limbal melanocytes, whereas conditioned medium from LEPC may support a dendritic, pigmentary phenotype.

Unlike melanocyte precursors in the limbal stroma, differentiated melanocytes in the limbal epithelium are constantly exposed to solar UV light. The exclusive expression of NGFR/p75^NTR^ in epithelial melanocytes and its induction by the limbal basement membrane component LN-511-E8 is consistent with its suggested role to prevent UV-induced apoptosis via upregulation of BCL-2 in epidermal melanocytes [31]. Nerve growth factor (NGF) secreted by neighboring keratinocytes has also been shown to induce dendrite formation, melanogenesis, and the migration of epidermal melanocytes [32]. Moreover, increased expression levels of CXCL12 (SDF-1) in epithelial melanocytes and their induction by LN-511-E8 in both melanocyte populations support the notion that CXCL12-CXCR4 signaling mediates the close association of limbal niche cells with LEPCs [33]. They further underpin the concept that limbal epithelial melanocytes play a critical role in immune regulation of the limbal stem cell niche [13]. Melanocyte-derived CXCL12 regulates the activation of melanocyte-specific immunity, inflammatory and immune responses, and recruitment of antigen-presenting and other immune cells [34]. Apart from these immunomodulatory functions, CXCL12 also regulates the migration, differentiation, and proper localization of immature melanocyte precursors [35].Accordingly, it has been previously shown that LN-511 can increase expression levels of CXCR4 in epidermal melanocytes [36].

These findings are partially consistent with previous studies reporting on the effects of keratinocyte-and fibroblast-derived soluble factors on epidermal melanocytes [37]. Keratinocyte-derived factors maintained melanocyte homeostasis by supporting melanocyte proliferation, melanogenesis, and survival. In parallel, paracrine factors derived from fibroblasts or mesenchymal cells were shown to either support or inhibit melanocyte proliferation and melanogenesis in a context-specific manner [37].

Taken together, the comparative analysis of two melanocyte populations suggests that stromal melanocytes may serve as a reservoir of bipolar immature precursor cells, which upon migration into the basal epithelial layer, assume a dendritic phenotype to tightly enwrap their associated LEPCs in the limbal niche [7,8]. Upon contact with the limbal basement membrane and paracrine factors from LEPCs, they appear to upregulate melanogenesis and expression of NGFR/p75^NTR^ and CXCL12 to promote increased resistance to UV-induced apoptosis and immunoregulatory functions. Re-distribution of ß-catenin from the nucleus to the cell membrane, where it is known to interact with E-cadherin to coordinate cell-cell adhesion, indicates inactive Wnt signaling in the limbal niche [38,39,40].

Our previous studies provided evidence that limbal epithelial melanocytes exert multiple functions in limbal niche homeostasis beyond UV protection of LEPCs [13]. In particular, limbal melanocytes were found to regulate the maintenance of LEPCs, support epithelial wound healing, control local immune reactions and angiogenesis. Based on their regulatory, immunomodulatory, and antio-angiogenic functions, limbal melanocytes hold great potential for regenerative therapies for ocular surface disorders.

Limbal melanocytes have been used successfully for improved tissue engineering of corneal epithelial cell sheets for ocular surface reconstruction [14,21,41]. For this purpose, they have been isolated from epithelial layers of organ-cultured human corneoscleral tissue [21,41]. However, a gradual depletion of epithelial cells from donor corneas has been demonstrated both after short-term hypothermic storage (>7 days) and long-term organ culture (>28 days), showing a marked loss of epithelial cells and extensive epithelial damage [42,43]. Despite its scarcity, well-preserved corneal tissue is, therefore, required to obtain and isolate sufficient numbers of melanocytes from the limbal epithelium. Identifying the presence of melanocytes in the anteriormost limbal stroma expands the options of isolating melanocytes from the limbal region, since the stromal tissue is usually well preserved even after long-term culture conditions. For example, it has been demonstrated that mesenchymal stem cells can be reliably isolated from the corneal stroma of organ-cultured corneas after long-term storage, attributed to their protected location within the stromal matrix [44]. Since the stromal melanocytes displayed similar functional properties to epithelial melanocytes, the stromal population may represent a valuable alternative resource for stem cell-based tissue engineering approaches. Although their anti-inflammatory and anti-angiogenic potential remains to be analyzed, their support of clonal growth of LEPCs suggests that they may enhance the long-term survival of LEPCs within corneal epithelial constructs similar to their epithelial equivalents [45].

The cultivation of primary melanocytes has been previously considered a challenge because of low cell numbers in tissues and low proliferation rates leading to frequent overgrowth by epithelial cells or fibroblasts in vitro [11,14,46]. Various methods have been suggested to establish pure melanocyte cultures from epidermal and limbal epithelia, with the most common ones relying on differential cytotoxic effects of geneticin on epithelial cells and fibroblasts [14,19,20,46]. At low concentrations, geneticin has very limited toxicity for slow-cycling melanocytes due to lower protein synthesis, but causes harm to actively synthesizing cells, particularly fibroblasts [20]. Although these procedures generate pure melanocyte populations, they are very time-consuming and potentially imply toxic side effects on the phenotype and function of the surviving melanocytes [22]. As confirmed by the present study, the cell viability of melanocytes exposed to geneticin for 24 h decreased significantly in a concentration-dependent manner. In addition, abundant cell debris resulting from damaged fibroblasts may adversely affect the proliferation and growth of the surviving melanocytes, which represent only about 1 to 5% of cells in epithelial and stromal cell suspensions.

To avoid these limitations, FACS technology was recently used for the direct isolation of melanocytes from epithelial cell suspensions based on the expression of surface markers like N-cadherin and CD117 (c-Kit) [11,22,23]. CD117 is expressed on both melanocytes and several stem cell populations, and binding to its ligand stem cell factor (SCF) plays an important role in stem cell and melanocyte homeostasis [47]. Since FACS is a rather sophisticated and expensive technique, mostly provided only at core facilities, we established an easier alternative method for the direct isolation and enrichment of limbal melanocytes from organ-cultured corneas by using readily available MACS technology. We preferentially used the negative selection protocol to eliminate contaminating epithelial cells and fibroblasts from epithelial and stromal melanocyte populations because the target cells were not conjugated to magnetic beads at any point in the process. We thus observed that MACS technology protected the target cells, produced more viable melanocytes with greater proliferative capacities, and shortened the culture time considerably from 3 months to 1 month. MACS technology has been previously applied for isolating LEPCs using αvß5 integrin as a positive marker and SSEA4 (stage-specific embryonic antigen-4) as a negative marker [48,49] but has not been used to isolate melanocytes. However, the simple and convenient protocol described in this study may offer an attractive tool for the isolation, cultivation, and exploration of these potent cell populations in future regenerative applications. Moreover, increased knowledge of these specialized cells might improve understanding of melanocyte-related diseases such as limbal melanoma and congenital aniridia [50].

## 4. Materials and Methods

### 4.1. Human Tissues and Study Approval

Human donor corneas with appropriate research consent were procured by the Erlangen Cornea Bank after corneal endothelial transplantation. Tissues were obtained after informed written consent from the relatives of the donors and used in accordance with the principles of the Declaration of Helsinki for experiments involving human tissues and samples. Ethics approval was obtained from the Institutional Review Board of the Medical Faculty of the University of Erlangen-Nürnberg (No. 4218-CH).

### 4.2. Melanocyte Isolation by Magnetic-Activated Cell Sorting (MACS)

Organ-cultured corneoscleral tissue with appropriate research content was provided by the Erlangen Cornea Bank after corneal endothelial transplantation. Corneoscleral buttons were cut into 12 one-clock-hour sectors, from which limbal segments were obtained by incisions made at 1 mm before and beyond the anatomical limbus. After digestion of limbal segments with 2.4 U/mL dispase II (Roche Diagnostics, Mannheim, Germany) at 37 °C for 1 h, epithelial sheets and stroma were separated by scraping with a spatula. The epithelial sheets were further dissociated into single cells by digestion with 0.25% trypsin and 0.02% EDTA (Thermo Fisher Scientific, Waltham, MA, USA) at 37 °C for 15 min. Single-cell suspensions were filtered through cell strainers with a pore size of 20 µm (Sysmex, Kobe, Japan) to eliminate remaining cell clumps or debris. In parallel, the remaining stroma was incubated with 2 mg/mL collagenase A (Roche Diagnostics, Mannheim, Germany) at 37 °C for 16 h, followed by filtration through cell strainers as described above to generate single-cell suspensions of limbal stroma.

Contaminating epithelial cells were depleted from the limbal epithelial cell suspension using CD326 (EpCAM) MicroBeads (Milteny Biotec, Bergisch Gladbach, Germany) according to the manufacturer’s instructions. After centrifugation at 300× *g* for 10 min, cells were resuspended in a binding buffer consisting of phosphate-buffered saline (PBS), 0.5% bovine serum albumin (BSA), and 2 mM EDTA. 5 × 10^7^ cells were then incubated in 300 µL binding buffer, 100 µL FcR Blocking Reagent, and 100 µL CD326 MicroBeads at 4 °C for 30 min. After washing with binding buffer and centrifugation at 300× *g* for 10 min, supernatants were aspirated to remove unbound microbeads. Cell pellets were resuspended in 500 µL binding buffer and passed through the magnetic field of a MACS cell sorting column (Milteny Biotec, Bergisch Gladbach, Germany). While positively labeled epithelial cells were retained in the column, the unlabeled melanocytes in the flow-through fraction were seeded into T25 flasks (Corning, Tewksbury, MA, USA) and cultivated in a CnT-40 melanocyte proliferation medium (CellnTEC, Bern, Switzerland) supplemented with 1 × penicillin–streptomycin–amphotericin B mix (Pan Biotech, Aidenbach, Germany) at 37 °C under 5% CO_2_ and 95% humidity. Media were changed every second day. After reaching about 60% confluency, cells were subjected to one or two additional MACS purification steps with Anti-Fibroblast MicroBeads (Milteny Biotec, Bergisch Gladbach, Germany) to obtain pure cultures of epithelial melanocytes.

Contaminating fibroblasts were depleted from the limbal stromal cell suspension using Anti-Fibroblast MicroBeads (Milteny Biotec, Bergisch Gladbach, Germany) based on a not further specified fibroblast-specific antigen. After centrifugation and resuspension, 1 × 10^7^ cells were incubated in 80 µL binding buffer and 20 µL Anti-Fibroblast MicroBeads at room temperature (RT) for 30 min. After washing with binding buffer, the unlabeled melanocytes were obtained by collecting the flow-through after magnetic sorting, seeded into T25 flasks, and cultivated as described above. After reaching about 60% confluency, cells were subjected to 2 to 3 additional MACS purification steps to obtain pure cultures of stromal melanocytes.

For a positive selection of melanocytes, epithelial and stromal cell suspensions were incubated with CD117 (c-Kit) MicroBeads (Milteny Biotec, Bergisch Gladbach, Germany) at 4 °C for 15 min. After washing two times, positively labeled melanocytes were eluted from the magnetic column by flushing with 1 mL of binding buffer. The enriched melanocytes were seeded into T25 flasks and cultivated as described above.

### 4.3. Melanocyte Isolation by Geneticin Treatment

Conventional geneticin treatment of melanocyte cultures was performed as previously described [21]. Briefly, single-cell suspensions obtained from limbal epithelial sheets and limbal stroma were seeded into T75 flasks and cultured in CnT-40 medium at 37 °C under 5% CO_2_ and 95% humidity. Contaminating epithelial cells were eliminated by differential trypsinization using 0.05% trypsin-0.01% EDTA (Thermo Fisher Scientific, Waltham, MA, USA) as described [21]. Contaminating fibroblasts were eliminated by 2 to 3 treatments with 0.2 mg/mL geneticin (G418; Thermo Fisher Scientific) in medium 254 (Thermo Fisher Scientific, Waltham, MA, USA) for 48 h as described [21].

### 4.4. Melanocyte Cultivation on Extracellular Matrix and in Conditioned Media

Recombinant laminin 511-E8 (LN-511-E8) (Nippi, Tokyo, Japan), representing the biologically active C-terminal portion of LN-511, was coated on 6-well plates (Falcon; Thermo Fisher Scientific, Waltham, MA, USA) at a concentration of 0.5 µg/cm^2^ as per manufacturers’ recommendations. Collagen type I (Sigma-Aldrich; St. Louis, MO, USA) was coated at a concentration of 6 μg/cm^2^ overnight at 4 °C. Epithelial and stromal melanocytes were seeded at a concentration of 1 × 10^5^ cells/well on pre-coated 6-well plates and cultivated for 72 h in CnT-40 medium. After reaching about 80% confluence, cells were harvested for RNA extraction.

Limbal epithelial stem/progenitor cells (LEPC) and limbal mesenchymal stromal cells (LMSC) were cultured in T75 flasks as previously described [13]. After reaching 80% confluence, media were changed to CnT-Prime Keratinocyte/Melanocyte Co-Culture medium (CellnTEC, Bern, Switzerland) or Mesencult medium (Stem Cell Technologies Germany; Köln, Germany), respectively. After 3 days of cultivation, conditioned media were collected, centrifuged at 1000× *g* for 10 min, and filtered through a 0.22 µm sterile filter (Millex; Merck, Darmstadt, Germany). Epithelial and stromal melanocytes were seeded at a concentration of 1 × 10^5^ cells/well in 6-well plates and cultivated in LEPC-or LMSC-conditioned media diluted in CnT-40 medium 1:9 for 72 h. After reaching about 80% confluence, cells were harvested for RNA extraction.

### 4.5. Cell Viability Assay (MTT)

Melanocytes were seeded in 96-well-plates at a density of 6 × 10^3^ cells per well. After cell adhesion, different concentrations (100 µg/mL, 200 µg/mL, 400 µg/mL) of geneticin in CnT-40 medium were added to the wells. After 24 h, 48 h, and 72 h incubation, 10 µL of 12 mM MTT stock solution (Thermo Fisher Scientific, Waltham, MA, USA) in 100 µL fresh CnT-40 medium was added and incubated for 4 h at 37 °C. Next, all but 25 µL of medium were removed from the wells before adding 50 μL of DMSO to each well and incubating for 10 min at 37 °C. Finally, absorbance was measured at 540 nm using a spectrophotometer (Multiskan Spectrum; Thermo Fisher Scientific, Waltham, MA, USA).

### 4.6. Flow Cytometry

Corneas were prepared as described above. Freshly isolated cells before MACS separation and cultivated epithelial and stromal melanocytes at passages 0 to 2 were characterized by flow cytometry using antibodies against CD117 (c-Kit) and Melan-A (Table 1). Cultures of pure melanocytes (P4) were used as a positive control, and unstained melanocytes and melanocytes stained with corresponding isotype antibodies were used as the negative control. All steps were performed at 4 °C. Before incubating cells with CD117-APC or isotype-APC antibodies (Milteny Biotec, Bergisch Gladbach, Germany) for 30 min in the dark, single-cell suspensions were blocked with FcR Blocking Reagent (Miltenyi Biotec, Bergisch Gladbach, Germany) for 5 min. Next, single-cell suspensions were fixed using a BD Fixation/Permeabilization solution kit (BD Biosciences; Heidelberg, Germany). After washing two times, cell suspensions were stained with Melan-A-FITC (Novus Biologicals; Littleton, CO, USA) or isotype-FITC antibodies (Miltenyi Biotec, Bergisch Gladbach, Germany) for 30 min. After two washing steps, cells were resuspended in 500 µL PBS and detected by LSRFortessa X-20 cell analyzer (BD Biosciences, Heidelberg, Germany) with FACS Diva Software 8.0.2. A total number of 10,000 events were acquired.

### 4.7. Immunohistochemistry and Immunocytochemistry

For immunohistochemistry of frozen sections, corneoscleral tissue samples obtained from 3 normal human donor eyes (mean age, 68.3 ± 7.6 years) within 15 h after death were embedded in optimal cutting temperature (OCT) compound and frozen in isopentane-cooled liquid nitrogen. Cryosections of 4 μm thickness were cut from the superior or inferior quadrants, fixed in cold acetone for 10min, blocked with 10% normal goat serum, and incubated in primary antibodies against Melan-A and integrin α6 (Table 1) diluted in PBS overnight at 4 °C. Antibody binding was detected by Alexa 488- and 555- conjugated secondary antibodies (Thermo Fisher Scientific, Waltham, MA, USA), and nuclear counterstaining was performed with DAPI (4,6-diamidino-2-phenylindole; Sigma-Aldrich, St. Louis, MO, USA). In negative control experiments, the primary antibodies were replaced by PBS or equimolar concentrations of an irrelevant isotypic primary antibody.

For immunocytochemistry, melanocytes were seeded on glass slides and fixed in methanol at −20 °C for 10 min. After air drying, the cells were permeabilized with 0.01% Tween 20 in PBS for 10 min and blocked with 0.5% cold-water fish gelatin and 0.1% ovalbumin (Sigma-Aldrich, St. Louis, MO, USA) in PBS for 45 min at RT. Primary antibody incubation was carried out overnight in blocking solution at 4 °C, and secondary antibodies were applied together with DAPI in PBS for 1 h at RT. After triple washing, slides were mounted in Aqua Poly Mount (Polysciences, Eppelheim, Germany). Immunolabelled cryosections and cultured cells were examined with a fluorescence microscope (BX51; Olympus, Hamburg, Germany) or a laser scanning confocal microscope (LSM 780; Carl Zeiss Microscopy, Oberkochen, Germany). Staining patterns were semi-quantitatively assessed in three different pairs of epithelial and stromal melanocyte populations on 10 microscopic fields covering the growth area (113 mm^2^) of glass slides at a magnification of ×40.

### 4.8. Quantitative RT-PCR

Total RNA of cultured melanocytes was extracted with the RNeasy Micro or Mini Kit (Qiagen, Hilden, Germany), including an on-column DNase digestion step according to the manufacturer’s instructions. Total RNA (0.5–1 μg) was reverse transcribed to cDNA using Superscript II reverse transcriptase (Invitrogen, Karlsruhe, Germany) as previously described [51]. PCR reaction was carried out in triplicate in a 1× reaction buffer SsoFast EvaGreen Supermix (Bio-Rad Laboratories, Feldkirchen, Germany) according to the manufacturers’ recommendations. Primer sequences (Eurofins, Anzing, Germany) are given in Table 2. For normalization of gene expression levels, ratios relative to the housekeeping gene GAPDH or ACTB (ß-actin) were calculated by the comparative CT method (ΔΔCT).

### 4.9. Cell Proliferation Assay

Cell proliferation was assessed with the cell proliferation ELISA BrdU kit (Roche Diagnostics, Mannheim, Germany) according to the manufacturer’s instructions. Briefly, 2 × 10^4^ melanocytes were seeded per well of 24-well plates and treated with LN-511-E8, collagen type I, or LEPC/LMSC-conditioned medium for 72 h as described above. Before fixation for 30 min at RT, BrdU labeling reagent was added at 1:1000 to each well and incubated overnight at 37 °C. Next, melanocytes were incubated with Anti-BrdU-POD working solution for 90 min at RT. After washing 3 times, substrate solution was added to each well for 5–30 min until sufficient color development. The reaction was stopped by 1 M H_2_SO_4_, and the absorbance was read at 450 nm using a spectrophotometer (Multiskan Spectrum; Thermo Fisher Scientific, Waltham, MA, USA).

### 4.10. Pigmentation Assay

The melanin content of melanocytes was measured as previously described [21]. Briefly, cell pellets of 1 × 10^5^ melanocytes were dissolved in 1 mL of 1N NaOH/10% dimethyl sulfoxide (DMSO) for 2 h at 80 °C. After centrifugation at 12,000× *g* for 10 min, the supernatants were transferred to 96-well plates with a standard curve (0–20 µg/mL) prepared from synthetic melanin (Merck, Darmstadt, Germany). The absorbance was read at 470 nm using a spectrophotometer, and the melanin concentration was calculated according to the standard curve.

### 4.11. Colony Forming Efficiency Assay

For colony-forming efficiency (CFE) assay, epithelial and stromal melanocytes were seeded into 6 well-plates at a density of 2 × 10^4^ cells/cm^2^. Single-cell suspensions of LEPC were prepared from limbal rims as described previously [13], seeded at a density of 1 × 10^3^ cells/cm^2^ on the melanocyte layers, and cultivated in CnT-Prime Keratinocyte/Melanocyte Co-Culture medium (CellnTEC, Bern, Switzerland) for approximately 14 days. Subsequently, the melanocytes were removed using Versene solution (Thermo Fisher Scientific, Waltham, MA, USA) for 30 s and rinsing under vigorous pipetting and microscopic control. After fixation with 4% paraformaldehyde for 1 h, the colonies were stained with 2% rhodamine B (Merck, Darmstadt, Germany) for 15 min. The CFE was calculated using the number of colonies formed divided by the number of cells plated × 100%. The colony growth area was calculated as colony growth area/total culture area × 100% using the ZEN blue imaging software (Carl Zeiss Microscopy, Oberkochen, Germany).

### 4.12. Statistical Analysis

Statistical analyses were performed using the GraphPad InStat statistical package for Windows (Version 8.3.0; GraphPad Software Inc., La Jolla, CA, USA). Data are reported as mean ± standard deviation (SD) from individual experiments. Group comparisons were performed using a ratio paired *t*-test. A *p*-value < 0.05 was considered statistically significant.

## Figures and Tables

**Figure 1 ijms-23-03756-f001:**
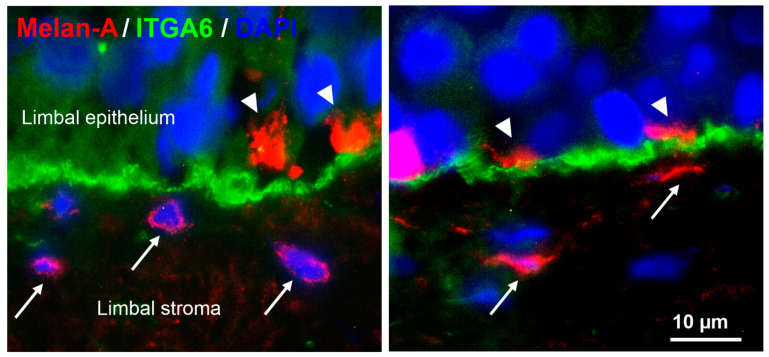
Localization of epithelial and stromal melanocytes at the human limbus in situ. Immunofluorescence double labeling of corneoscleral tissue sections prepared from two donor corneas (age 65 and 72 years) showing Melan-A^+^ melanocyte processes (arrowheads) in the basal limbal epithelium expressing integrin α6 (ITGA6) and additional Melan-A^+^ melanocytes (arrows) in the subepithelial limbal stroma; nuclear counterstaining with 4′,6-diamidino-2-phenylindole (DAPI).

**Figure 2 ijms-23-03756-f002:**
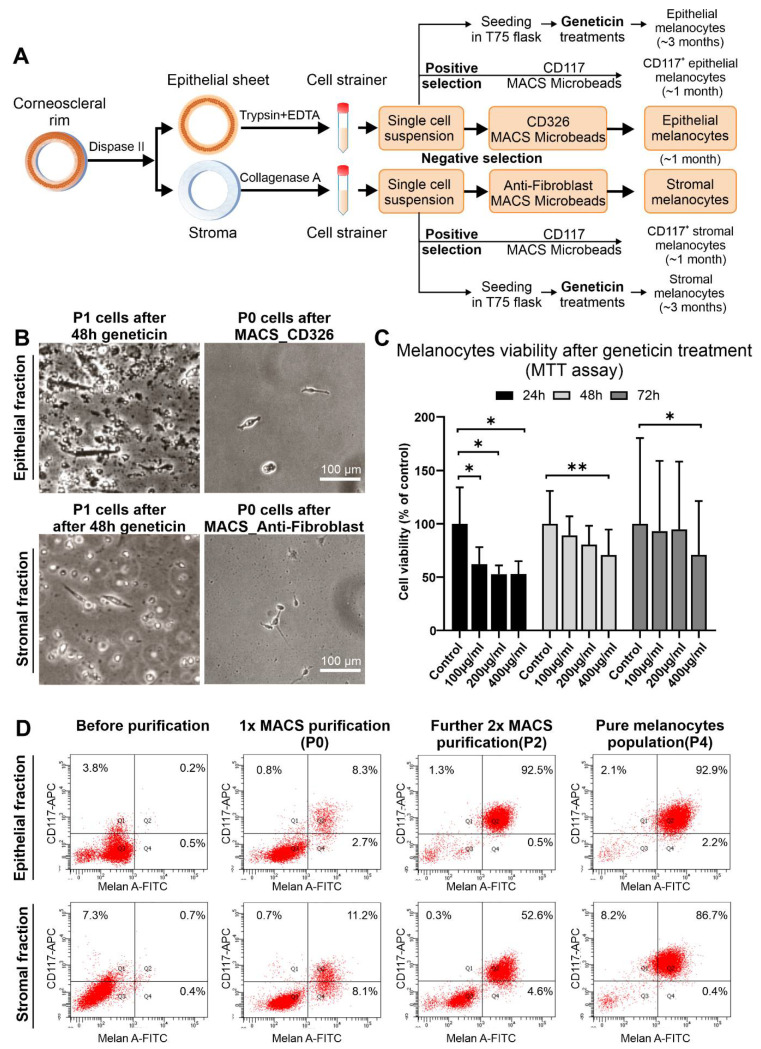
Isolation of limbal epithelial and stromal melanocyte populations using conventional geneticin treatment and magnetic-activated cell sorting. (**A**). Graphical representation of limbal melanocyte isolation: Differential enzymatic digestion of limbal tissue generated single-cell suspensions of epithelial and stromal cells; melanocytes were enriched from both suspensions either by conventional geneticin treatments or by magnetic-activated cell sorting (MACS) technology using either a positive selection approach with CD117 (c-Kit) Microbeads or a negative selection approach with CD326 (EpCAM) Microbeads and Anti-Fibroblast Microbeads, respectively. (**B**). Phase-contrast images showing epithelial (**B**) and stromal (**C**) cell fractions after 48 h of geneticin treatment at passage 1 (P1) and after MACS negative selection at passage 0 (P0). (**C**). Cell viability (MTT) assay after 24, 48, and 72 h of treatment with different concentrations of geneticin (100-400 µg/mL). Data (% of control) are expressed as means ± SD (*n* = 3) relative to control (* *p* < 0.05; ** *p* < 0.01; paired *t*-test). (**D**). Flow cytometry analysis showing an increase of Melan-A^+^/CD117^+^ cells in epithelial and stromal fractions after MACS purification steps.

**Figure 3 ijms-23-03756-f003:**
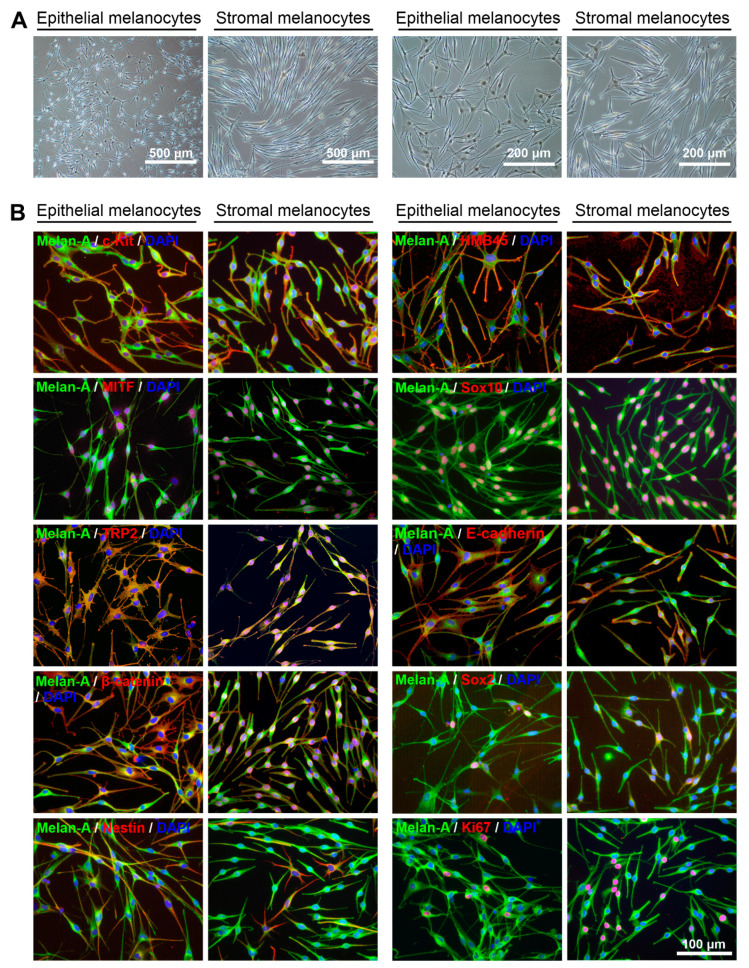
Phenotypic and immunocytochemical characterization of limbal epithelial and stromal melanocytes. (**A**). Phase-contrast images showing pure populations of epithelial and stromal melanocytes at passage 4 at two different magnifications. (**B**). Immunofluorescence double labeling of epithelial and stromal melanocytes for Melan-A and additional melanocyte-associated markers c-Kit (CD117), HMB45 (PMEL, premelanosome protein), MITF (microphthalmia-associated transcription factor), Sox10 (SRY-box transcription factor 10), TRP2 (tyrosine-related protein 2), E-cadherin, ß-catenin, Sox2, nestin, and Ki-67; nuclear counterstaining with 4′,6-diamidino-2-phenylindole (DAPI). The magnification bar at the bottom-right corner applies to all images.

**Figure 4 ijms-23-03756-f004:**
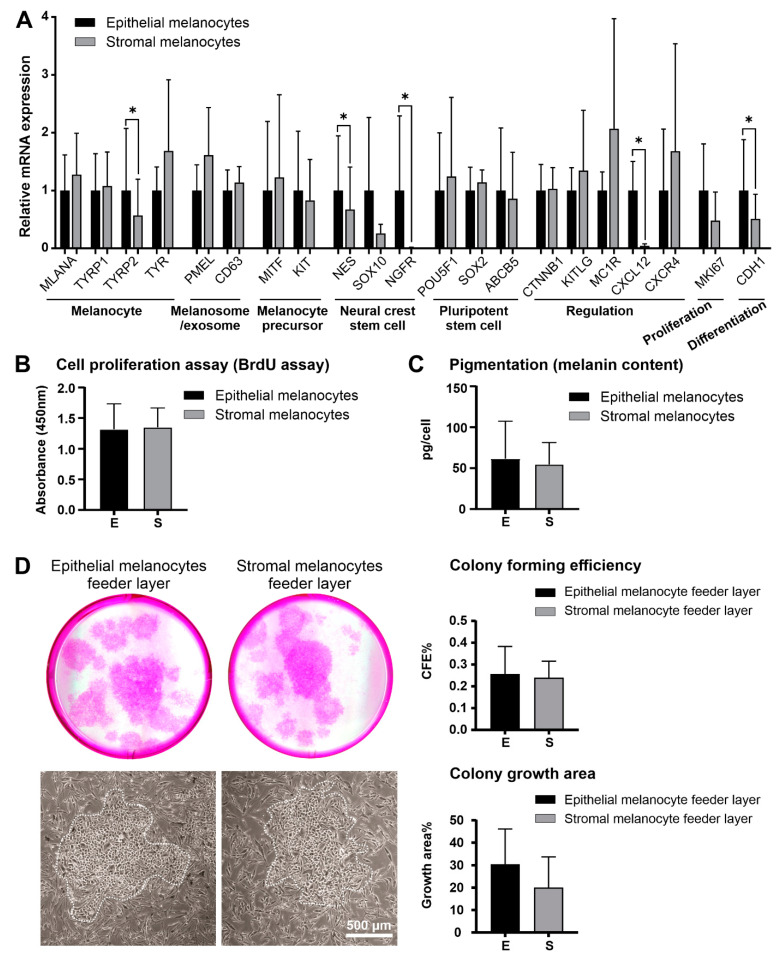
Comparative analysis of limbal epithelial and stromal melanocytes. (**A**). Quantitative real-time polymerase chain reaction (qRT-PCR) primer assays showing relative expression levels of MLANA (Melan-A), TYRP1 and TYRP2 (tyrosinase-related protein 1 and 2), TYR (tyrosinase), PMEL (premelanosome protein), CD63 (CD63 antigen), MITF (microphthalmia-associated transcription factor), KIT (c-Kit/CD117), NES (nestin), SOX10 (SRY-Box transcription factor 10), NGFR (nerve growth factor receptor), POU5F1 (POU class 5 homeobox 1), SOX2 (SRY-Box transcription factor 2), ABCB5 (ATP binding cassette subfamily B member 5), CTNNB1 (catenin beta 1), KITLG (Kit ligand), MC1R (melanocortin 1 receptor), CXCL12 (C-X-C Motif Chemokine Ligand 12), CXCR4 (stromal cell-derived factor 1 receptor), MKI67 (Ki-67), and CDH1 (E-cadherin) in limbal epithelial and stromal melanocytes. Data are normalized to GAPDH and ACTB and expressed as means (2^−ΔCT^ × 1000) ± SD (*n* = 5) relative to epithelial melanocytes (* *p* < 0.05; paired *t*-test). (**B**). Proliferative capacity of limbal epithelial and stromal melanocytes as determined by BrdU incorporation assay. Data are expressed as means ± SD (*n* = 5). (**C**). Quantification of cellular melanin content in limbal epithelial and stromal melanocytes. Data are expressed as means ± SD (*n* = 5). (**D**). Formation of limbal epithelial stem cell clones (dotted lines) expanded on feeder layers of limbal epithelial and stromal melanocytes, followed by quantitative comparative analysis of colony-forming efficiency (CFE) and colony growth area. Data represent means ± SD (*n* = 3).

**Figure 5 ijms-23-03756-f005:**
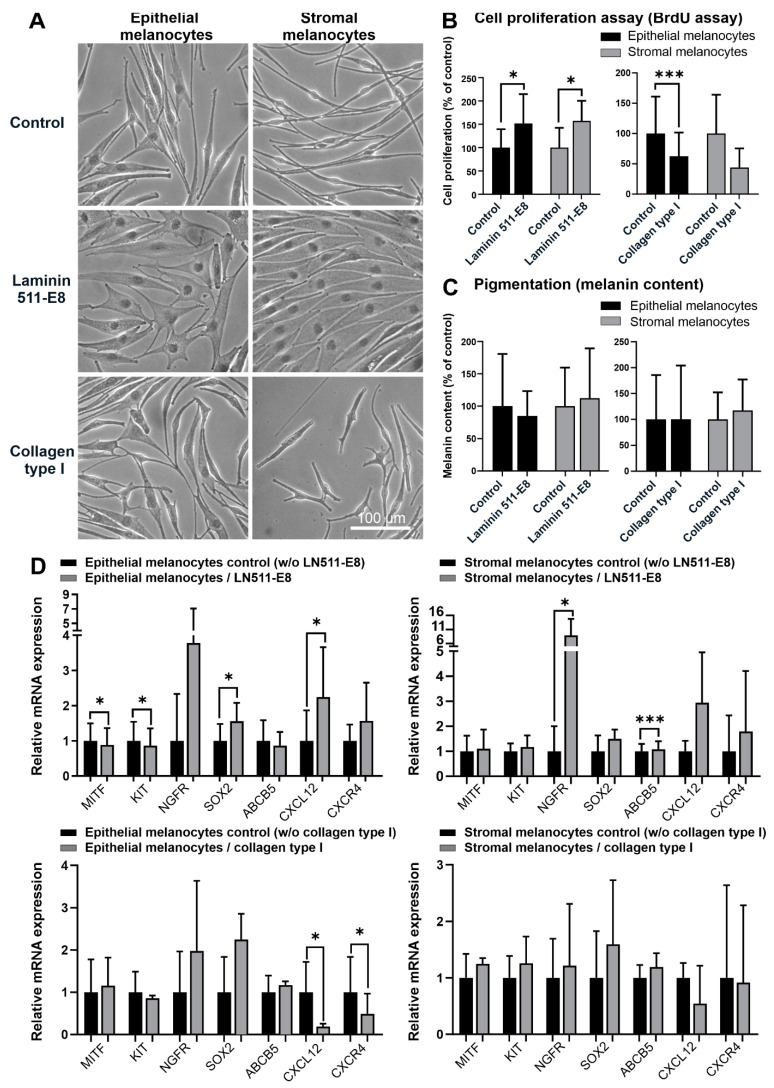
Effects of extracellular matrix components on limbal epithelial and stromal melanocytes. (**A**). Phase-contrast images of limbal epithelial and stromal melanocytes cultured on laminin 511-E8 and collagen type I (the magnification bar at the bottom-right corner applies to all images). (**B**). The effect of laminin 511-E8 and collagen type I on the proliferation of limbal epithelial and stromal melanocytes was analyzed by BrdU incorporation assay after 72 h. Data are expressed as means ± SD (*n* = 4) relative to control (* *p* < 0.05; *** *p* < 0.001; paired *t*-test). (**C**). The effect of laminin 511-E8 and collagen type I on the melanin content of limbal epithelial and stromal melanocytes was analyzed by pigmentation assay after 72 h. Data are expressed as means ± SD (*n* = 4) relative to control. (**D**). Quantitative real-time polymerase chain reaction (qRT-PCR) primer assays showing relative expression levels of MITF (microphthalmia-associated transcription factor), KIT (c-Kit/CD117), NGFR (nerve growth factor receptor), SOX2 (SRY-Box transcription factor 2), ABCB5 (ATP binding cassette subfamily B member 5), CXCL12 (C-X-C Motif Chemokine Ligand 12), and CXCR4 (stromal cell-derived factor 1 receptor) in limbal epithelial and stromal melanocytes cultured on laminin 511-E8 and collagen type I. Data are normalized to GAPDH and ACTB and expressed as means (2^−ΔCT^ × 1000) ± SD (*n* = 3) relative to control (* *p* < 0.05; *** *p* < 0.001; paired *t*-test).

**Figure 6 ijms-23-03756-f006:**
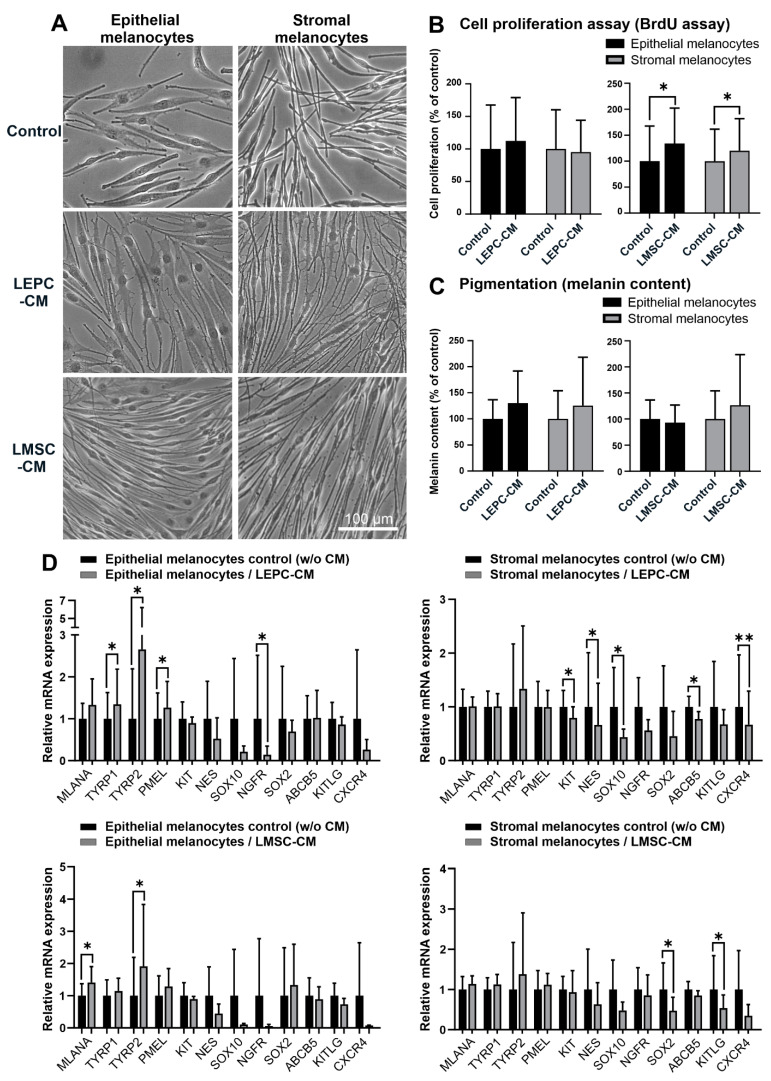
Effects of soluble factors on limbal epithelial and stromal melanocytes. (**A**). Phase-contrast images of limbal epithelial and stromal melanocytes cultured in conditioned media derived from limbal epithelial stem/progenitor cells (LEPC-CM) and limbal mesenchymal stem cells (LMSC-CM) (the magnification bar at the bottom-right corner applies to all images). (**B**). The effect of LEPC-CM and LMSC-CM on the proliferation of limbal epithelial and stromal melanocytes was analyzed by BrdU incorporation assay after 72 h. Data are expressed as means ± SD (*n* = 4) relative to control (* *p* < 0.05; paired *t*-test). (**C**). The effect of LEPC-CM and LMSC-CM on the melanin content of limbal epithelial and stromal melanocytes was analyzed by pigmentation assay after 72 h. Data are expressed as means ± SD (*n* = 4) relative to control. (**D**). Quantitative real-time polymerase chain reaction (qRT-PCR) primer assays showing relative expression levels of MLANA (Melan-A), TYRP1 and TYRP2 (tyrosinase-related protein 1 and 2), PMEL (premelanosome protein), KIT (c-Kit/CD117), NES (nestin), SOX10 (SRY-Box transcription factor 10), NGFR (nerve growth factor receptor), SOX2 (SRY-Box transcription factor 2), ABCB5 (ATP binding cassette subfamily B member 5), KITLG (Kit ligand), and CXCR4 (stromal cell-derived factor 1 receptor) in limbal epithelial and stromal melanocytes cultured in LEPC-CM and LMSC-CM. Data are normalized to GAPDH and ACTB and expressed as means (2^−ΔCT^ × 1000) ± SD (*n*
*=* 4) relative to control (* *p* < 0.05; ** *p* < 0.01; paired *t*-test).

**Table 1 ijms-23-03756-t001:** List of antibodies used for immunofluorescence.

Antibody (Clone)	Host Species	Source	Concentration	Application
Isotype control antibody, APC	mouse	Miltenyi Biotec	1:50	Flow cytometry analysis
Isotype control antibody, FITC	mouse	Miltenyi Biotec	1:50	Flow cytometry analysis
CD117 antibody, anti-human, APC	mouse	Miltenyi Biotec	1:20	Flow cytometry analysis
Melan-A/MART-1 antibody, FITC	mouse	Novus Biologicals	1:133	Flow cytometry analysis
Melan-A	rabbit	Abcam	1:500	Immunofluorescence
Melan-A (A103)	mouse	DAKO	1:25	Immunofluorescence
Integrin α6 (ITGA6) (GoH3)	rat	Millipore	1:100	Immunofluorescence
CD117/c-Kit (Ab81)	mouse	Cell signaling	1:200	Immunofluorescence
Sox10 (BC34)	mouse	Abcam	1:50	Immunofluorescence
TRP2/DCT	rabbit	Invitrogen	1:100	Immunofluorescence
β-catenin (15B8)	mouse	Thermo Fisher	1:100	Immunofluorescence
E-cadherin (36)	mouse	BD	1:100	Immunofluorescence
Nestin (10C2)	mouse	Abcam	1:100	Immunofluorescence
P75 NGFR (NGFR5)	mouse	Abcam	1:100	Immunofluorescence
Ki67 (MIB-1)	mouse	DAKO	1:100	Immunofluorescence
MITF (C5)	mouse	Abcam	1:150	Immunofluorescence
Sox2	rabbit	Cell signaling	1:100	Immunofluorescence
HMB45 (HMB45)	mouse	DAKO	1:50	Immunofluorescence
TRP1 (TA99)	mouse	Abcam	1:50	Immunofluorescence

**Table 2 ijms-23-03756-t002:** Primers used in qRT-PCR primer assays.

Gene Symbol	Accession Number	Sequence	
ABCB5	NM_001163941.2	5	CAGCAAGGGAAGCAAATGCGTA
		3	ATCCTCTGTTTCTGCCCTCCAC
ACTB	NM_001101.3	5	GCGTACAGGTCTTTGCGGATG
		3	TGAGGCACTCTTCCAGCCTTC
CD63	NM_001780.6	5	TACGTCCTCCTGCTGGCCTTTT
		3	ATGACCACTGGCAACAGAGAGC
CDH1	NM_001317184.2	5	CAGGATGGCTGAAGGTGACAGA
		3	AGAGCACCTTCCATGACAGACC
CTNNB1	NM_001904.4	5	GGTCTGAGGAGCAGCTTCAGT
		3	GGCCATGTCCAACTCCATCAAA
CXCL12	NM_000609.7	5	CATGCCGATTCTTCGAAAGCCA
		3	CTGTTGTTGTTCTTCAGCCGGG
CXCR4	NM_001008540.2	5	GACCGCTTCTACCCCAATGACT
		3	TACCAGGCAGGATAAGGCCAAC
GAPDH	NM_002046.2	5	AAGGTCGGAGTCAACGGATTTGG
		3	ATGACAAGCTTCCCGTTCTCAGC
KIT	NM_000222.3	5	AAGCACAATGGCACGGTTGAAT
		3	ACAGGGTGTGGGGATGGATTTG
KITLG	NM_003994.5	5	ACCTTGTGGAGTGCGTGAAAGA
		3	AGTAAAGAGCCTGGGTTCTGGG
MC1R	NM_002386.4	5	CATCTCTGACGGGCTCTTCCTC
		3	AGGCAGCAGATGAAGCAGTACA
MITF	NM_198159.3	5	CGGGCTCTGTTCTCACTTTCCA
		3	GAGCTTATCGGAGGCTTGGAGG
MKI67	NM_002417.5	5	TACGGATTATACCTGGCCTTCCC
		3	ACAACAGGAAGCTGGATACGGA
MLANA	NM_005511.2	5	TTACTGCTCATCGGCTGTTGGT
		3	AGACACTTTGCTGTCCCGATGA
NES	NM_006617.2	5	CAGCGTTGGAACAGAGGTTGGA
		3	TCTGTAGGCCCTGTTTCTCCTG
NGFR	NM_002507.4	5	CACCGACAACCTCATCCCTGT
		3	GCTGTTGGCTCCTTGCTTGTTC
PMEL	NM_001200053.1	5	GTCAGCACCCAGCTTATCATGC
		3	CTGCTATGTGGCAACTGGGGTA
POU5F1	NM_002701.6	5	GGAGAAGGAGAAGCTGGAGCAA
		3	GCAGATGGTCGTTTGGCTGAAT
SOX2	NM_003106.4	5	GAAGGATAAGTACACGCTGCCC
		3	CGTTCATGTGCGCGTAACTGTC
SOX10	NM_006941.4	5	CATCCAGGCCCACTACAAGAGC
		3	TCTGTCTTCGGGGTGGTTGGAG
TYR	NM_000372.5	5	TCATCCAAAGATCTGGGCTATGAC
		3	GACGACACAGCAAGCTCACA
TYRP1	NM_000550.3	5	GTCGCTCAGTGCTTGGAAGTTG
		3	GTCATACTTTCCCGTGGGGTCA
TYRP2/DCT	NM_001129889.2 Variante 2	5	GGACAAACGCTTTGCCACATTC
		3	GAAGGGAGTTCCTTGGTCGCTT

## Data Availability

Not applicable.

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
