# Peer review of "Identification, Isolation, and Characterization of Melanocyte Precursor Cells in the Human Limbal Stroma"

_ijms, 2022, doi:10.3390/ijms23073756_

Round 1

Reviewer 1 Report

This paper describes a newer approach to isolating limbal melanocytes and the identification of a stromal melanocyte precursor population from donor human limbus. The use of MACS over FACS is a variation on a theme and provides a lower cost alternative to the use of FACS to isolate these cells.  A more direct comparison of FACS and the plating technique to MACS on proliferation and gene expression would make this more impactful.

Comments

  1. There were some formatting issues that made following the narrative difficult at times. Lines 137-142 are missing.
  2. In figure 3A what is the difference between the two sets of images? If it is just magnification this should be indicated. In fact, the magnification of all images should be indicated.
  3. In figure 3B, C-kit, Sox2 and nestin expression not convincing. There seems to be a low proportion of cells that are positive for these proteins and those that are appear to be negative for Melan-A.
  4. What is the basis of the conclusions in figure 3, is it counting cells? How many fields? How many replicates?
  5. In figure 4, how was the colony growth area calculated?
  6. In lines 220-233 the discussion of the RT-PCR data, results should be limited to only those with a significant change.
  7. Better images in Figure 6A would make the dendriform processes more visible, currently it is difficult to see a difference between LEPC_CM and LMSC-CM.

Reviewer 2 Report

Dear authors,

The authors are reported the identification and characterization of melanocyte precursors, what were isolated from human limbal stromal. The study is well written and interesting for the scientific community, to understand the cell renewal and healing of the limbus and the cornea.

I have some comments:

Major:

  • Could the authors confirm the significant difference in Fig 2 and 4, because when the variation of the samples is so big, the readers might have some questions?
  • To explain why Geneticin is used, the authors could refer the Publication PMID 6724622
  • The authors isolated the cells from elderly (over 61 years old). Could the authors discuss in a short paragraph the potential age effect on proliferation rate, CFE, etc?
  • in Line 287, the authors mentioned "To the best of our knowledge, this is the first report showing the presence of subepithelial melanocytes in the anteriormost subepithelial stroma at the human limbus." I was wondering if the publication PMID: 33067486 was not the first publication showing the presences of melanocytes in the anteriormost subepithelial stroma?

Minor:

  • Sometimes the font is changing over the manuscript and it should be corrected.
  • All latin words must be in Italic.
  • line 153-154. There is a presentation problem. 

Round 2

Reviewer 2 Report

Dear Authors,

I have no additional comments.

Sincerely